# LFPS: Learned Farthest Point Sampling

## Abstract

The processing of point clouds with deep neural networks is relevant for many applications, including remote sensing and autonomous driving with LiDAR sensors. To ensure the computational feasibility of point cloud processing, it is crucial to reduce the cloud's resolution, i.e., its number of points. This downsampling of point clouds requires a deep learning model to abstract information, enabling it to process points within a more holistic context. A traditional technique for reducing the resolution of a point cloud is Farthest Point Sampling (FPS). It achieves a uniform point distribution but does not adapt to the network's learning process. In contrast, learned sampling methods are adaptive to the network but cannot be seamlessly incorporated into diverse network architectures and do not guarantee uniformity. Thus, they can miss informative regions of the point cloud, reducing their effectiveness for large-scale point cloud applications.

To address these limitations and bridge the gap between algorithmic and learned sampling methods, we introduce Learned Farthest Point Sampling (LFPS), an innovative approach that combines the advantages of both algorithmic and learned techniques. Our method relies on a novel loss function designed to enforce a uniform point distribution. We show by theoretical proof that its minima guarantee a uniformity comparable to FPS. Furthermore, we extend the loss function to include information about key points, enabling the network to adaptively influence point selection while preserving uniform distribution in relevant as well as less relevant regions. In experimental studies, we evaluate the performance of LFPS both independently and within existing network architectures. Our results (a) show that LFPS serves as a plug-in alternative for algorithmic sampling methods, particularly as a faster alternative to FPS for large-scale point clouds, and (b) confirm the enhanced performance of LFPS across various tasks, emphasizing its versatility and effectiveness.

## 1 Introduction

With the expanding use of point clouds generated by sensors across a wide range of applications, there is growing demand for and interest in developing methodologies that effectively address the unique challenges posed by these datasets. One key method is downsampling, which plays a critical role in various applications. It is an essential component of numerous network architectures (Qian et al., 2022; Qi et al., 2017b; Fang et al., 2024). By reducing computational complexity and resource demands, downsampling not only accelerates processing times for large-scale point data but also facilitates the extraction of higher-level features. In the context of machine learning, downsampling is a fundamental component of network architectures. In 2D computer vision, pooling and strided convolutions iteratively summarize large image areas into smaller feature maps, improving both model efficiency and performance by providing a more holistic view of the data. The analogous principle applies to point-based machine learning networks, where a downsampling method needs to determine which points to retain as the network progresses through its layers.

Current methods for point-based downsampling can be categorized into standalone algorithmic and learnable sampling methods. Our objective is to synergize the strengths of both groups by introducing our universally applicable Learned Farthest Point Sampling (LFPS) method which can be integrated into neural networks that require a downsampling of points. Learned sampling approaches cannot guarantee full coverage of the entire point cloud, which can be especially problematic for large-scale point clouds as visible in Fig. 1. In contrast, LFPS achieves a uniform point distribu-

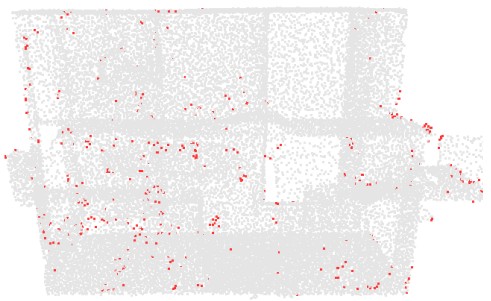 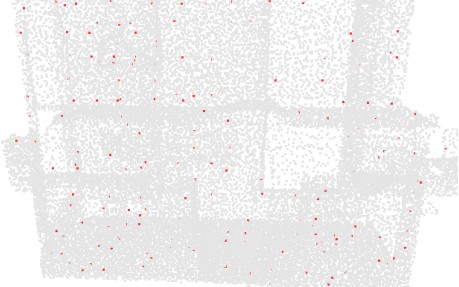

Figure 1: Learned sampling with APES (left) compared to LFPS (right). LFPS covers the entire point cloud while focusing on informative regions, resulting in improved performance.

tion that resembles Farthest Point Sampling (FPS) while retaining the flexibility to prioritize more informative regions of the point cloud. Our main contributions can be summarized as follows:

- Formulation of a novel loss function derived from the sampling properties of FPS, along with a theoretical proof that the loss function's minima correspond to a sampling scheme with FPS-equivalent characteristics.

- Development of a refined framework that leverages learned sampling, allowing the network to influence point selection.

- Extensive ablation studies demonstrating that a network trained with the proposed loss function can attain the predicted minima and illustrating the effectiveness of the underlying operating principles.

- Experimental evidence showing that LFPS can serve as a seamless replacement of existing approaches in supervised and unsupervised learning models resulting in improved performance. Notably, experiments on large-scale point clouds demonstrate improved runtime efficiency compared to FPS and enhanced accuracy compared to a learned sampling method.

## 2 POINT CLOUD SAMPLING IN DEEP LEARNING APPLICATIONS

There are two primary categories of sampling methods for point clouds. The first category consists of task-agnostic algorithmic approaches, which aim to sample points either uniformly or with high efficiency. The second category comprises learned sampling methods, where the selection of points is based on the network's preferences and task-specific requirements. The most established algorithmic sampling method is *Farthest Point Sampling (FPS)*, which ensures that the sampled points are evenly distributed across the point cloud (Eldar et al., 1997). This is achieved by iteratively selecting the point farthest from the already chosen points. However, FPS is sensitive to outliers, computationally inefficient for large-scale point clouds, and lacks permutation invariance, as the results depend on the initial starting point. Of course, FPS is not the only task-agnostic algorithmic method; it has some competitors. *Grid sampling*, a faster alternative, generates new points based on the distribution of points within cells of a predefined grid (Wu et al., 2022). Despite its speed advantage, grid sampling is also sensitive to outliers, can lead to less uniform distributions, and is constrained by the grid structure. *Random sampling*, a conceptually simple method, which is, e.g., used by Hu et al. (2019), can exacerbate density imbalances and overlook important points within the cloud.

Those algorithmic sampling methods have been widely integrated into various network architectures. The pioneering point-based network *PointNet* (Qi et al., 2017a) processes point clouds in a single hierarchy, while its successor, *PointNet++* (Qi et al., 2017b), learns hierarchical local features across multiple layers and downsampling stages, employing FPS for its downsampling operations. *KPConv* (Thomas et al., 2019) uses a set of learnable kernel points to adaptively process point clouds

and relies on grid sampling to control input point density, doubling the grid size for downsampling. Recently, transformer-based architectures like *Point Cloud Transformer (PCT)* (Guo et al., 2021) have gained popularity. The basic transformer architecture computes global attention based on pairwise relationships between all input tokens, significantly increasing memory and computational costs. *Point Transformer* (Engel et al., 2020; Wu et al., 2022) reduces this complexity by applying local attention to neighboring points, with grid sampling reducing input size. Unsupervised approaches, such as *Point-M2AE* (Zhang et al., 2022), a masked autoencoder with hierarchy, and in-context learning methods (Fang et al., 2024), also utilize FPS for downsampling.

Learned sampling methods, which represent the second major group of sampling approaches, take a different approach. Dovrat et al. (2019) were among the first to apply deep learning for point cloud sampling, proposing *S-Net*, which generates a simplified point cloud optimized for a downstream task. However, the simplified output is not necessarily a subset of the original point cloud, requiring post-processing to match each simplified point to its nearest neighbor. *SampleNet* (Lang et al., 2020) addresses this issue by introducing a differentiable relaxation of the matching operation. Yang et al. (2019) leveraged *Gumbel Softmax* to modify the sampling behavior during training and inference. The *Critical Points Layer* (Nezhadarya et al., 2020) offers a permutation-invariant sampling technique that retains key points based on the maximum feature values produced. *APSNet* (Ye et al., 2022) uses attention-based sampling with a simplified PointNet and an LSTM to select the most informative points, jointly optimizing sampling and task loss during training on point cloud videos. *APES* (Wu et al., 2023) is an attention-based method designed for sampling points along the edges of a point cloud. Meanwhile, Wang et al. (2023) propose a transformer-based sampling technique *LighTN* aimed at improving efficiency. *ADS* (Hong et al., 2023), on the other hand, clusters points with mean shift clustering before selecting the most informative ones from each cluster. Additionally, Wen et al. (2023) present a method that preserves object geometry by generating a skeleton as prior knowledge and using it to guide the sampling process. Despite these advancements, task-adaptive sampling methods face significant challenges when integrated into deep network architectures as replacements for algorithmic sampling methods. Directly differentiable downsampling methods, such as S-Net, SampleNet, and LighTN, demonstrate their value in obtaining an initial simplified point cloud that can be processed more efficiently by existing networks. However, these methods cannot be seamlessly integrated into network structures because the features at higher levels are typically not derived from point positions, but depend on the features of the previous layer, making meaningful gradient computation for point positions unattainable. Furthermore, methods like ADS and APES, which rely on point features from earlier layers, are unsuitable for architectures, where sampling occurs before feature computation, such as in Point-M2AE. Additionally, none of the learned sampling methods explicitly guarantee a uniform distribution of sampled points, which can lead to significant information loss, particularly in large-scale point clouds. In all these scenarios, LFPS provides an effective solution to address these challenges.

## 3 LEARNING TO SAMPLE FARTHEST POINTS

Below, we first revisit the key properties of FPS and leverage the insights to derive a novel loss function for training a data-driven sampling method. Finally, we use a theoretical sketch to show that minimizing this loss function leads to a distribution that maintains FPS' uniformity guarantees.

### 3.1 CHARACTERIZING FARTHEST POINT SAMPLING

FPS is an iterative procedure that starts from an arbitrary initial point and subsequently selects points that are maximally distant from the set of already chosen points. This approach is designed by Eldar et al. (1997) to ensure a relatively even spread of points by maximizing the minimum distance to the nearest neighbor at each step. While FPS does not necessarily optimize for the mean nearest neighbor distance or minimize its variance, it does mitigate aliasing artifacts common in overly regular sampling patterns, particularly in its original context, i.e., image processing. In the context of point cloud data, given a sufficiently large number of points, this results in similar nearest neighbor distances for all points, approximating an optimal uniform distribution (see Section 4).

To analyze the uniformity of the distribution of points sampled by FPS, denoted as $S_{\text{FPS}}$, a *Voronoi diagram (VD)* is employed. In this framework, points are considered neighbors if, and only if, they share an edge in the VD. The properties of FPS are formalized with two distance measures:

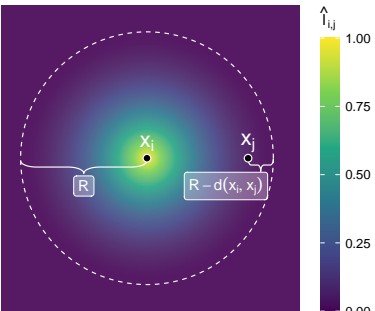 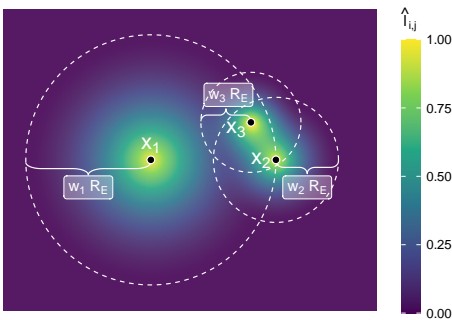

Figure 2: The key variable $R$ represents the radius around a selected point, within which a newly placed point incurs a loss value of $\hat{l}_{i,j}$.

Figure 3: Weighted sampling enables the use of individually adjusted $R_E$ values and similarity measures $\hat{l}_{i,j}$ for each point, allowing for sampling with variable density.

$R_M$, the maximum distance from a sample point in $S_{\text{FPS}}$ to any vertex of the VD, and $R_m$, the minimum distance from a sample point to a vertex of the VD. The following theoretical bounds were established by Eldar et al. (1997):

- For any set of points $S_{\text{FPS}}$ the inequality $R_M \leq 2 \cdot R_m$ holds.
- The pairwise distance between any two points $s_i, s_j \in S_{\text{FPS}}$ is at least $R_M$.
- The distance between any two neighboring points in $S_{\text{FPS}}$ is no more than $2 \cdot R_M$.

### 3.2 A Loss Function to Emulate Farthest Point Sampling

Given those distance bounds, we propose a loss function, denoted as $\mathcal{L}_{\text{LFPS}}(S)$, to evaluate a sampled set $S$ by considering each point and its associated neighbor relationships:

$$\mathcal{L}_{\text{LFPS}}(S) = \frac{1}{|S|} \sum_{i \in S} l(x_i, N_i^S), \tag{1}$$

where $N_i^S$ represents the set of neighbors of point $x_i$ within $S$. We first examine the desired properties of $l(x_i, N_i^S)$ in the continuous case, where points are a direct output of the network's computation, and then describe an approach to transfer the loss to the discrete case, where the model can only select points from a given discrete set $P$. The function $l(x_i, N_i^S)$ should satisfy two key conditions: First, it should be higher if the average distance to the neighbors is relatively small compared to the average neighbor distances of the other points. Second, it should reach its minimum when all neighbors are at distance $R$ so that $R = 2 \cdot R_m = 2 \cdot R_M$ is maximized. Note that $R$ is not known but can be estimated. The first condition can be formalized by defining a similarity measure as the negative distance between $x_i$ and $x_j$. To satisfy the second condition, we set the similarity measure to zero for distances equal to or greater than $R$. This results in the expression $\max(R - d(x_i, x_j), 0)$. See Fig. 2 for a visual depiction of these parameters. Furthermore, to penalize points that are very close to $x_i$, we square the similarity measure. Finally, to remain dataset-agnostic, this bounded similarity can be normalized to be in $[0, 1]$ by dividing by $R^2$. Thus, a possible choice for $l(x_i, N_i^S)$ is

$$l(x_i, N_i^S) = \sum_{j \in N_i^S} \hat{l}_{i,j} \quad \text{where } \hat{l}_{i,j} = \max\left\{1 - \frac{1}{R} \cdot d(x_j, x_i), 0\right\}^2. \tag{2}$$

However, this loss function can only guide networks that have a direct influence on the position of the points, meaning when the coordinates are a direct output of the network's computation. In the case of discrete point positions that are to be selected, $l$ can only be applied implicitly. Instead, the network can compute scores for each point and subsequently select the points with the highest scores. Consequently, we derive a loss function by combining the similarity measure with these

predicted scores. Therefore, suppose that there are $n = |P|$ points in the point cloud, each assigned a predicted score $s_i$, from which to sample $\lceil n/f_d \rceil = |S|$ points. Here, $P$ denotes the set of points available for sampling and $f_d$ denotes the decrease factor. Further, let $N_i \subset P$ denote the neighborhood of $x_i$ in the initial point cloud, i.e., before downsampling. Then, for a given point $x_i \in S$, there exist $k < n$ nearest neighbor points $x_j \in N_i$. Note that in the discrete case we use $k$-nearest neighbor relationships instead of neighbor relationships in the context of Voronoi diagrams, as they are easier to compute and perform better in batches. Assuming the score values lie between 0 and 1 and the $|S|$ points with the highest scores are to be selected, we propose the following loss function that effectively distributes the points throughout the $k$-nearest neighbor graph of $P$ such that the number of $k$-nearest neighbors with a high score in each selected point's neighborhood is minimized.

$$l(x_i, N_i) = \frac{1}{k+1} \left( (1-s_i)^2 + \sum_{j \in N_i} s_j^2 \right). \tag{3}$$

The second term is essentially the mean squared error (MSE) for the scores assigned to the neighbors of the selected points, where the error quantifies the extent to which these neighbors are also selected; ideally, they should not be selected at all and thus should receive a score of $0$. This MSE would be trivially zero if the network predicts a score of $0$ for every point, leading to the selection of points, depending only on the tie breaker rule of the $\max$ function. The first term counteracts such a trivial solution by enforcing the score of the selected points to be $1$. However, this does not lead to a uniform distribution of the selected points when the points in $P$ are not already uniformly distributed, meaning the nearest neighbor distances between the points in $P$ are not all equal. To achieve favorable selections in this case as well, the approach for a loss function from Eq. (2) is combined with that from Eq. (3). By weighting each neighbor's score according to the similarity $\hat{l}_{i,j}$ of the neighboring point, closer selected neighbors exert a greater influence on the loss function, regardless of their ordering. Specifically, a selected neighbor outside the $R$ radius does not increase the loss function, while in dense regions, the loss is generally higher. Therefore, a network trained using this loss function implicitly learns to select points distant from each other. Using $\hat{l}$ from Eq. (2), this leads to

$$l(x_i, N_i) = \frac{1}{k+1} \left( (1-s_i)^2 + \sum_{j \in N_i} \left( s_j \cdot \hat{l}_{i,j} \right)^2 \right). \tag{4}$$

Notice again that the loss – as indicated in Eq. (2) – is calculated based on the $k$-nearest neighborhood of $x_i$ within $P$, not within $S$. This distinction does not affect the minimum of Eq. (2), as non-selected points should be assigned a score of $0$ by the network. This approach allows more points to be included in the loss function calculation, ensuring that they receive a gradient signal.

From a practical standpoint, the choice of the unknown distance $R$ depending on different datasets poses a challenge. Initial experiments suggest estimating $R$ as $R_E$, defined as the 1st quartile of the $k$-nearest neighbor distance for points $x_i \in P$, which ensures resilience to outliers. The parameter $k$ should be chosen such that $R_E$ slightly overestimates $R$ (for an appropriate choice of $k$, see Section 4.1.2). Although this will lead to a non-zero loss function, there exists a range of values for $R_E > R$ where the theoretical properties of FPS are still satisfied in a minimum of the loss, as stated in the following theorem.

**Theorem** For a task in which $n$ points are to be selected from a bounded $\mathcal{R}^2$ region defined as $\{(x,y) \in \mathcal{R}^2 : a \leq x \leq b, c \leq y \leq d\}$, let $S$ denote the set of selected points. Given that the maximum attainable first nearest neighbor distance is $R$, and with an estimated optimal distance $R_E = R + \varepsilon$, there exists $\varepsilon > 0$ such that the distribution of points in $S$ that minimizes the loss function $\mathcal{L}_{LFPS}$ exhibits the same distance properties as FPS in the two-dimensional continuous case.

**Proof Sketch** In Appendix A, we provide the detailed proof for the existence of a range of values for $R_E$ such that minimizing the loss function results in a specific distribution of selected points. This proof applies to the two-dimensional continuous case, following the proof of the FPS properties. While this result is derived for a continuous domain, it remains valuable for point clouds that approximate a manifold. The key idea is to relate the problem of finding the point configuration

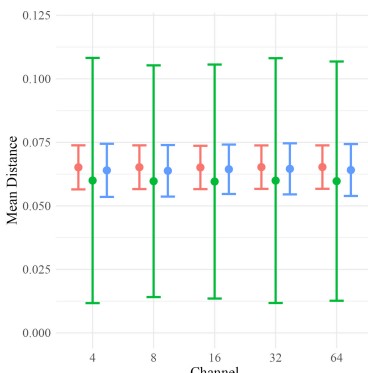 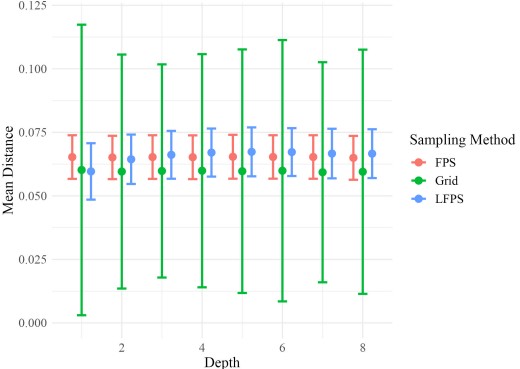

Figure 4: Point distribution obtained by varying the number of channels (left) and number of layers (right) in the selection network for LFPS (blue), compared to FPS (red) and grid sampling (green). Points represent the mean distance, and error bars indicate the corresponding standard deviation. For LFPS, 3 layers and 16 channels are sufficient to achieve performance comparable to FPS.

with minimal loss to the problem of optimal circle packing in two dimensions. The loss reaches exactly zero, i.e., its minimum, if no point lies within the radius $R_E$ of any other point. Therefore, by drawing a circumcircle with radius $R_E/2$ around each point, it is required that no two circles intersect. This condition makes an optimal configuration equivalent to a solution of the circle packing problem. If $R = R_E$, the optimal packing of points in a continuous space is achieved through a hexagonal lattice structure, as this corresponds to the optimal solution to the circle packing problem (Thue, 1892). In this case, the loss function reaches its minimum value (zero), which reproduces the distributional properties of FPS. Now, consider the cases where $R \neq R_E$: If $R > R_E$ the loss function can be minimized through multiple configurations, as the corresponding circles do not need to be tightly packed, leading to non-unique point distributions. This is undesirable as it may break the specific properties of FPS, particularly when $R_E$ is too small. If $R < R_E$ the loss function cannot reach zero, as no valid configuration allows all points to maintain the desired spacing. For any configuration that deviates from the optimal hexagonal circle packing, there must be at least one pair of points positioned closer together than the optimal distance. This further implies that multiple pairs of points are spaced farther apart due to the squaring of the similarity measure. To formalize this, we derive a loose upper bound on the possible reduction of the loss for any configuration other than the hexagonal packing. We consider a packing for which $R_E$ is sufficiently close to $R$ such that all farther-spaced point pairs can again be in hexagonal packing. This upper bound is inserted into the loss inequality, allowing us to compute the fraction of the reduced distance between the closest point pair for which the inequality holds. It shows that the inequality only holds for values that fulfill $R_M \leq 2 \cdot R_m$. Thus, there must exist a range of values for $R_E$ such that the desired properties of FPS are preserved, even with a loose bound.

### 3.3 WEIGHTED SAMPLING

To harness the advantages of learned sampling, the loss function can be expanded to guide the network in sampling regions of interest more densely than others. The regions of interest can be any point-to-importance assignment defined by the user. For example, points with labels that are commonly misclassified in semantic segmentation, or more generally, points with higher activations, can be assigned higher importance values. These importance values are only required during loss computation and can be decoupled from the actual selection process in the network. Consequently, even activations from later layers in architectures such as u-net (Ronneberger et al., 2015) can be utilized to compute importance values, and the selection network must learn to predict which points will be valuable in subsequent processing steps. This decoupling allows our task-adaptive sampling strategy to be integrated into architectures, e.g. Point-M2AE, where feature information is unavailable at the time of point sampling, in contrast to most other sampling techniques. Let each point $x_i$ be assigned an importance value $v_i \in [0, 1]$, where 1 denotes high importance and 0 signifies low importance. While points with higher importance should be sampled with higher probability, the overall sampling should maintain an even distribution among points with a similar importance.

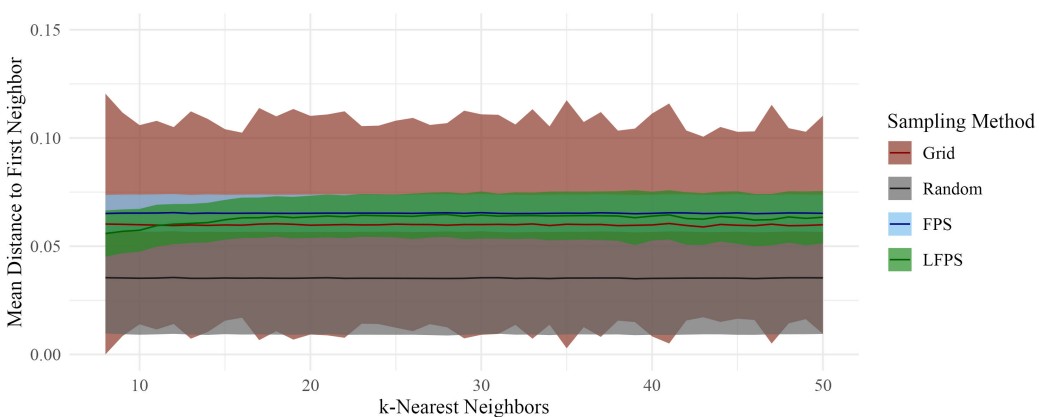

Figure 5: Development of the mean and variance of the first nearest neighbor distance across various neighborhood sizes $k$ compared with three other algorithmic sampling methods. LFPS achieves its best performance for $24 \leq k \leq 32$.

We introduce two methods to influence sampling based on importance values, one updating $R_E$ per point and the other one the neighbor distances. Both methods utilize a weighting function $w_i = p_u - (p_u - p_l) \cdot v_i$. The user-defined parameters $p_u$ and $p_l$ determine the extent of influence, with $p_u$ setting the upper bound for the weight and $p_l$ the lower bound, so $p_u = p_l$ implies no influence. In the first method, individual distances are obtained per point with $w_i \cdot R_E$. This enables the network to densely pack important points without incurring a loss for points that are too close to each other (see Fig. 3). However, the per-point loss function may increase for unimportant points that have a selected important point in their neighborhood, leading to a penalty for selecting this crucial point. To address this, the second method adjusts the neighbor similarities starting from Eq. (2) with $\hat{l}_{i,j} \cdot w_j$ in the neighborhood of a chosen point. Incorporating these adjustments into the loss computation yields the overall loss function, where $R$ in Eq. (2) for $\hat{l}$ is replaced by $R_E \cdot w_i$:

$$\mathcal{L}_{LFPS}(S) = \frac{1}{|S| \cdot (k+1)} \sum_{i \in S} \left( (1 - s_i)^2 + \sum_{j \in N_i} \left( s_j \cdot w_j \cdot \hat{l}_{i,j} \right)^2 \right), \tag{5}$$

## 4 EXPERIMENTS

While a loss function that ensures an even distribution of selected points is desirable, it does not guarantee that a network trained with this loss will converge to such a minimum. Therefore, we conduct several ablation studies to analyze the behavior of the standalone LFPS module. Subsequently, we test it within modern deep learning architectures to demonstrate the advantages of learned, well-distributed sampling. The core structure of our LFPS module is a compact ResNet (He et al., 2016) characterized by varying layer depths $d_R$ and channels per layer $c_R$. In each layer, the point itself, along with its $k$-nearest neighbors, is processed. A final layer predicts scores for each point.

### 4.1 STANDALONE LEARNED FARTHEST POINT SAMPLING

We explore various network configurations in an effort to identify settings that can produce a well-distributed sampling. These configurations are compared against the distribution of sampled points generated by FPS, grid and random sampling. The learned sampling strategy is trained for $2\,000$ steps, each with a batch size of $256$.

### 4.1.1 ESSENTIAL FEATURES FOR LEARNING A UNIFORM DISTRIBUTION

The first configuration involves testing the influence of input information using the S3DIS dataset (Armeni et al., 2016), with large indoor point cloud scenes. The experiment is conducted with

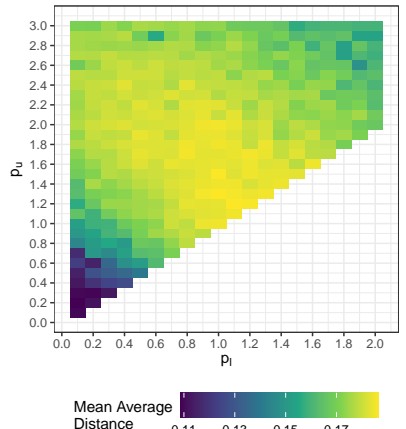 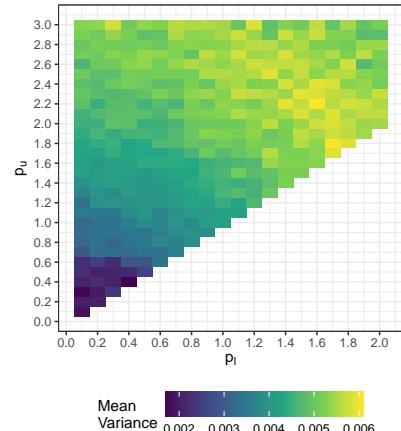

Figure 6: Varying the upper ($p_u$) and lower ($p_l$) bounds of the weighted sampling in a selection task for Point-M2AE. The mean nearest neighbor distance and variance of the selected points for each combination of $p_u$ and $p_l$ are presented. Excessively high values lead to increased variance, while overly low values prevent the network from learning an effective configuration.

fixed parameters $d_R = 2, c_R = 16, k = 24, f_d = 4, n = 2\,000$, and $n = 2\,000$, while varying the input information: either using point position information alone or combining point position information with distance information to the nearest neighbors. The results demonstrate that without the distance information, the network's performance does not surpass that of random sampling. Additionally, we examine the impact of the selection network's depth and width (i.e., the number of channels), specifically analyzing the mean nearest neighbor distance and its variance (see Fig. 4). The findings indicate limited improvement in selection quality beyond three consecutive selection layers, while the number of channels shows only a minor effect, with optimal performance observed at 16 channels.

### 4.1.2 INFLUENCE OF THE DISTANCE PARAMETER

The second analysis investigates the impact of the chosen $R_E$ on the sampling performance and examines how well the theoretically predicted range of possible values for $R_E$ aligns with the values observed empirically. To this end, we test the important parameter $k$, which also determines $R_E$ in the loss function, while keeping the initial analysis configuration fixed at $d_R = 2$ and $c_R = 16$, and varying $k$ in the range from 8 to 50. As illustrated in Fig. 5 and theoretically predicted, there is a range for $k$ (approximately 24 to 32) where the distribution of FPS is most closely approximated. Similar results are observed for other datasets (see Appendix B).

### 4.1.3 IMPACT OF THE WEIGHTING PARAMETERS

Third, we investigate the effect of the weighted sampling parameters $p_u$ and $p_l$, while keeping the remainder of the initial configuration fixed. For this experiment, we employ pre-sampled point clouds obtained by FPS from the ModelNet dataset. For the point-to-importance assignment, we use the sum of the normalized activations per point from the upward pass of a fully trained Point-M2AE model. We test all combinations of $p_l$ from 0.1 to 2.0 and $p_u$ from $p_l$ to 3.0 in increments of 0.1, whereas setting $p_u < p_l$ would reverse the importance of the points. After $2\,000$ training steps, we report the average mean and variance of the first nearest neighbor distances of the selected points over the last 100 steps (see heat maps in Fig. 6). The theoretically predicted range for the continuous two-dimensional setting is visible along the diagonal, where $p_u = p_l$, meaning that the selection model does not differentiate between important and unimportant points. Excessively high values lead to an increased variance in nearest neighbor distances, while overly low values result in the model failing to learn the task effectively. The optimal configuration should aim for a moderate variance to avoid large gaps in the point cloud, and maintain an adequate average distance to prevent redundant information from overly similar points. In general, the greater the difference between

| PC | LFPS | FPS | Grid |
|----|------|-----|------|

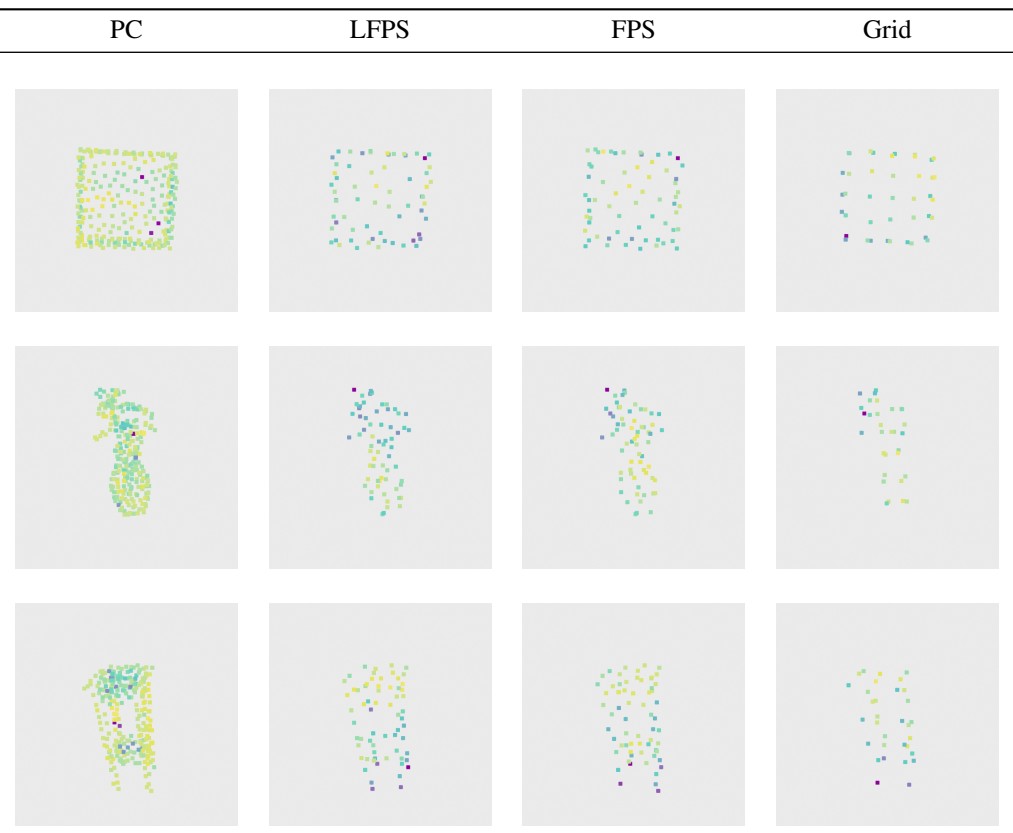

Figure 7: Qualitative comparison of different sampling strategies. The left column shows the point cloud (PC) color-coded by the cumulative activations each point receives from Point-M2AE. For each sampling method, brighter colors indicate denser regions. While FPS and LFPS initially appear similar, subtle differences emerge in regions with high and low activations. Specifically, in the last row, FPS sampled numerous points between the stool's legs – an area of low importance – whereas LFPS effectively avoided these unnecessary points.

$p_u$ and $p_l$, the more the model is compelled to discern which points are important. Based on these observations, we suggest setting $p_u = 2.4$ and $p_l = 2.2$.

### 4.1.4 TIME COMPLEXITY

In addition to the adaptability of LFPS compared to FPS, our method demonstrates significantly improved computational efficiency for large-scale point clouds. Specifically, the time complexity of LFPS is $\mathcal{O}(n)$, excluding the $k$-nearest neighbor computations, which are generally required for network operations regardless. Even when including the $k$-nearest neighbor computations and employing an optimized CUDA implementation of FPS, the execution time for sampling $25\,000$ points from $100\,000$ points is approximately 5 seconds for FPS, whereas LFPS achieves this in only 46 milliseconds.

### 4.2 REPLACING ALGORITHMIC SAMPLING METHODS IN MODERN NETWORK ARCHITECTURES

To demonstrate the performance of LFPS, we integrate it into recent deep learning architectures by replacing grid sampling in Point Transformer V2 and FPS in Point-M2AE, both in the last sampling layer. Since simpler sampling tasks are expected to show that LFPS outperforms FPS but not necessarily other learned sampling methods, this complex setting offers a more compelling evaluation of our model's performance. Our approach demonstrates its true advantages

Table 1: Performance ((m)IoU in %) of the Point Transformer V2 experiments on the S3DIS dataset, comparing grid sampling (PTv2), LFPS (PTv2$_{\text{LFPS}}$) and APES (PTv2$_{\text{APES}}$).

| model | all | ceiling | floor | wall | column | window | door | table | chair | sofa | bookcase | board | clutter |
|---|---|---|---|---|---|---|---|---|---|---|---|---|---|
| PTv2 | 68.3 | 91.8 | 98.5 | **85.8** | 28.8 | 60.6 | 71.5 | 81.4 | 92.1 | 63.0 | 75.2 | **83.4** | 55.7 |
| PTv2$_{\text{LFPS}}$ | **70.2** | **92.7** | 98.5 | 84.5 | 33.0 | **60.8** | **80.8** | **82.3** | **92.3** | **72.3** | **77.0** | 79.4 | **59.3** |
| PTv2$_{\text{APES}}$ | 63.2 | 89.9 | **98.6** | 81.3 | **44.8** | 56.8 | 49.3 | 77.3 | 88.9 | 52.3 | 67.5 | 65.3 | 49.6 |

in scenarios where other sampling strategies are not integrable or struggle to handle challenging point cloud properties, such as large-scale data. To take advantage of the potential improvements from weighted sampling, we use the activations from the network's upward pass, assigning higher importance to points with stronger activations. We set the sampling network parameters as $d_R = 3, c_R = 32, k = 32, p_u = 2.4, p_l = 2.2$, and compare our results against those obtained by the reference implementation.

We evaluate three versions of Point Transformer V2 on the S3DIS dataset: the original architecture (PTv2), one incorporating LFPS (PTv2$_{\text{LFPS}}$), and another utilizing the initially introduced sampling method APES (PTv2$_{\text{APES}}$), which can be directly integrated into the transformer architecture. The results can be seen in Table 1. PTv2$_{\text{LFPS}}$ improves the performance of the sophisticated network from 68.3 % mean intersection over union (mIoU) to 70.2 % mIoU. Notably, the performance on challenging object classes, such as column, door, and sofa, has improved due to the preferential sampling of informative points. As shown in Fig. 1, the APES module in PTv2$_{\text{APES}}$ tends to over-sample high-interest points, while entirely disregarding regions of the point cloud with lower scores. Consequently, this leads to a significant decrease in performance, especially for flat classes such as door and board, which lack many edge points and are therefore detected with lower accuracy.

Point-M2AE is trained in an unsupervised manner on ShapeNet (Chang et al., 2015), and its performance is evaluated based on the expressiveness of the codeword using a linear SVM on ModelNet. Replacing FPS with LFPS in this network improves accuracy, even on this challenging unsupervised learning task, to 92.8 %, compared to 92.4 % achieved by the reference implementation. For a qualitative comparison of the subtle yet meaningful differences in point selection between FPS and LFPS in Point-M2AE, as well as the notable differences between LFPS, grid, and random sampling, see Fig. 7. This comparison illustrates how our selection process can highlight important regions, while maintaining a similar point cloud coverage to FPS.

## 5 CONCLUSION

In this paper, we introduced LFPS, based on the first density-aware sampling loss function that harmonizes the strengths of traditional algorithmic sampling with the adaptability of learned techniques for point cloud processing. LFPS thus addresses the shortcomings of existing methods by integrating the uniformity of FPS with the data-specific learning capability of deep networks, ensuring balanced and efficient point selection. Our approach is grounded in a rigorous theoretical framework that establishes its similarity to FPS while LFPS is still able to focus on important regions. LFPS was tested within two existing network architectures, namely Point-M2AE, Point Transformer V2 serving as exemplars for a broad range of applications, demonstrating seamless integration and substantial improvements in runtime and accuracy. Additionally, LFPS proves highly effective for large-scale point cloud tasks, improving both computational efficiency and performance. Beyond improving runtime and accuracy, LFPS demonstrates the potential to generalize across different domains and tasks, including supervised and unsupervised learning, making it a versatile tool for further advancements in point cloud processing and broader 3D data applications.

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
