# OpenReview forum: "LFPS: Learned Farthest Point Sampling"
_ICLR.cc/2025/Conference — ICLR 2025 Conference Withdrawn Submission_

### Official Review · Reviewer_XB9h · 2024-10-16

**Soundness:** 3
**Presentation:** 4
**Contribution:** 4
**Rating:** 8
**Confidence:** 3

**Summary:**

This work introduces a learnable sampling strategy, LFPS with a well-designed loss function. LFPS can sample the points uniformly and effectively. The work is effective in supervised and unsupervised tasks, especially large-scale point cloud tasks.

**Strengths:**

1. The description of the proposed method is clear and easy to follow. The work starts by characterizing the FPS, then proposes an initial version of the loss function (eq. (1)), and gradually improves the loss function step by step with a detailed explanation.
2. The work also provides theoretical proof that makes the work more solid in mathematics.
3. The proposed work is also effective and efficient considering the experimental results.

**Weaknesses:**

1. LFPS looks like a combination of standalone algorithmic and learnable sampling methods. Is there any reason that LFPS is only compared with FPS? It would make the experiments more solid if LFPS can be compared with other learnable sampling methods, like LighTN, to make the experiments more solid.

**Questions:**

1. For figure 1, why does the total number of red points look different for the two images? When comparing the point distribution after sampling, it would be more fair that the total number of red points is the same.
2. For line 399, is there a typo that n=2000 is defined twice.

---

### Official Review · Reviewer_5B9f · 2024-10-29

**Soundness:** 2
**Presentation:** 2
**Contribution:** 2
**Rating:** 3
**Confidence:** 3

**Summary:**

This paper introduces a learning-based point cloud downsampling method, namely the learned farthest point sampling (LFPS) method. This method aims to balance the advantages of traditional algorithm sampling techniques (such as farthest point sampling) and adaptive learning sampling methods. In order to solve the problem that traditional methods are difficult to adapt to the network learning process, the authors introduce a new loss function that forces uniform point distribution, so that LFPS strives to ensure uniform coverage comparable to FPS, while allowing the network to adaptively prioritize key points in the point cloud. The authors tried to verify the effectiveness of these methods from both theoretical and experimental aspects. The learning-based adaptive LFPS was verified in the point cloud semantic segmentation algorithm of the downstream task.

**Strengths:**

The proposed LFPS was evaluated on two existing point-cloud architectures, Point-M2AE and Point Transformer V2, which serve as examples for a wide range of applications.

LFPS was experimentally compared on large-scale point cloud tasks to show the computational efficiency and performance.

The paper also presents a visual comparison of point cloud processing across different domains and tasks, such as supervised and unsupervised learning methods.

**Weaknesses:**

The explanation of the designed loss function should be improved, for example, two conditions related to the function $l(x_i, N_i^S)$ lacks clarity.

In the experiment, results for baseline methods such as random sampling or those with and without distance information are either missing or not clearly presented. A more thorough comparison with these methods would strengthen the claims of improvement.

The parameter settings for the uniform distribution experiments (e.g., distance metrics, k-nearest neighbors, and fixed points n=2000) are very specific and lack justification.

The time complexity analysis lacks comprehensive coverage of all experimental setups. Specifically, there is no breakdown of time results. A more detailed analysis would provide better insight into the relative time efficiency of LFPS against a range of sampling methods.

**Questions:**

The explanation of the two conditions for the function $l(x_i, N_i^S)$ is not easily understandable. It would be easier to follow if the authors can visualise two cases of associated neighbour relationships to explain these two conditions.

In the experiment of learning uniform distribution Sec. 4.1.1 or Fig. 4, where are the results of random sampling method and the methods with/without using distance information presented? Apart from the results, why is the result of using distance information better than the network with only point position information?

In Sec. 4.1.1, why are the parameters of the experiments for uniform distribution selected very specifically? For example, the distance, k-nearest neighbor, the number of points n = 2000.

What is the effect of the number of channels and why does the method with 16 channels have the optimal performance observed in the experiment?

About the experiment of time complexity, how are the results of all experiments, for example FPS with/without using knn or optimized CUDA. How is the time of these FPS and LFPS distributed? Apart from FPS, is there any comparison with other previous efficient sampling methods?

In Table 1, the performance of the two methods are only performed by using one model Point Transformer V2. How about the generalization of the proposed method? How about the comparison of the efficiency in this task?

---

### Official Review · Reviewer_EDfj · 2024-11-01

**Soundness:** 3
**Presentation:** 3
**Contribution:** 3
**Rating:** 5
**Confidence:** 5

**Summary:**

This paper proposes an innovative approach that combines the advantages of both algorithmic and learned techniques in downsampling point clouds. The method relies on a novel loss function designed to enforce a uniform point distribution. The authors also prove the effectiveness of the proposed method both theoretically and experimentally.

**Strengths:**

There are some spotlights in this paper:
* The proposed LFPS combines the advantages of both algorithmic and learned sampling methods at the same time.
* The authors provide detailed proof of the proposed loss function.
* Compared with FPS, LFPS not only achieves better performance but also efficiency.

**Weaknesses:**

I didn't find critical drawbacks to the proposed method. However, there are still some unclearnesses that need to be clarified.
* How to define the neighbors in equation 1?
* LFPS is extremely similar to FPS in Figure 7.
* Experiments on more datasets are expected. It would be great if the authors could conduct experiments on 2 more datasets.
* Experiments on more state-of-the-art models are expected. It would be great if the authors could conduct experiments with 3 more sota models.
* Experiments on more tasks are expected. It would be great if the authors could conduct experiments on 2 more tasks.
* Experiments on more sampling methods are expected. Table 1 showcases that LFPS is better than griding sampling and APES. Compared to FPS in Point-M2AE, the performance gain of LFPS is marginal.

**Questions:**

My questions are listed in the weaknesses section. If the author can clarify the unclearnesses, I will consider improving my rating.

**Details Of Ethics Concerns:**

No concerns.

---

### Official Review · Reviewer_uGqZ · 2024-11-03

**Soundness:** 2
**Presentation:** 2
**Contribution:** 3
**Rating:** 3
**Confidence:** 4

**Summary:**

1. This paper introduces Learned Farthest Point Sampling (LFPS), a method for processing point clouds with deep neural networks. The authors identify limitations in traditional Farthest Point Sampling (FPS), which achieves uniform distribution but lacks adaptability, and in learned sampling methods, which are adaptive but may overlook important regions.
2. LFPS aims to combine the strengths of both approaches through a novel loss function that enforces uniformity while allowing adaptive point selection. The paper includes theoretical proof indicating that LFPS can achieve uniformity comparable to FPS. Experimental results are presented to evaluate LFPS's performance, showing it as a faster alternative to FPS for large-scale point clouds.
3. This work contributes to the field by addressing the challenges in point cloud downsampling, though further validation across diverse networks and tasks may be beneficial.

**Strengths:**

1. The question posed in this paper—how to design an advanced point cloud sampling method that combines the efficiency of independent algorithms with the flexibility of learnable sampling algorithms—is indeed a meaningful research direction.

2. In contrast to papers without theoretical analysis, the authors provide some theoretical analysis to prove their ideas.

**Weaknesses:**

1. Lack of Novelty: The contributions of the paper do not present significant new insights or ideas compared to existing literature. It would be beneficial to clearly delineate how this work differentiates itself from prior research in point cloud sampling algorithms.

2. Insufficient Experiments: The experiments only include semantic segmentation. There is a lack of common point cloud downstream tasks such as classification, detection, and part segmentation. Additionally, experiments on various classic point cloud processing networks, such as PointNet++, PointNeXt, PointMLP, PointMAE, PointM2AE, and Point Transformer V3, are missing, leading to an inability to demonstrate the generalizability of the algorithm.

3. Theoretical Proof: Most of it is derived from the original FPS paper, which provides theoretical proof in a two-dimensional context, including the design of Voronoi diagrams and the theoretical aspects in section 3.1. It remains to be proven whether these theories applicable in 2D space can be extended to 3D space. In fact, two-dimensional space and three-dimensional space are likely very different, so the relevant conclusions or assumptions used by the author in the manuscript may not hold in three-dimensional space.

4. Lack of Necessary Illustrations: The paper lacks some essential graphical explanations, such as the design of Voronoi diagrams in section 3.1 and the specific implementation of the algorithm in section 3.2, which results in poor readability.

5. Unclear Logical Structure: The logical flow of the writing is not very clear. For instance, in the experimental section of chapter four, it is customary to first highlight the algorithm's performance on point cloud downstream tasks before discussing the ablation experiments regarding parameter selection.

**Questions:**

1. Theoretical Analysis: Could you provide a corresponding proof of the algorithm's applicability in three-dimensional space, addressing the theoretical analysis concerns mentioned earlier?

2. Choice of Comparison Methods: The experimental phase includes a limited number of sampling methods for comparison, specifically only FPS and APES. More tasks and architectural experiments can provide evidence of the method's effectiveness and generality.

---

### Author Response · Authors · 2024-11-15

We would like to express our gratitude to all reviewers for taking the time to engage with our work. However, after careful consideration of the feedback and scores received, we have decided to withdraw our paper.

---

### Note · Authors · 2024-11-15

I have read and agree with the venue's withdrawal policy on behalf of myself and my co-authors.